# Individualized Dosage of Tacrolimus for Renal Transplantation Patients Based on Pharmacometabonomics

**DOI:** 10.3390/molecules27113517

**Published:** 2022-05-30

**Authors:** Xiaoying He, Xi Yang, Xiaoting Yan, Mingzhu Huang, Zheng Xiang, Yan Lou

**Affiliations:** 1Zhejiang Provincial Key Laboratory for Drug Clinical Research and Evaluation, Department of Clinical Pharmacy, The First Affiliated Hospital, College of Medicine, Zhejiang University, 79 QingChun Road, Hangzhou 310000, China; hxy17858505013@163.com (X.H.); 13735514968@163.com (X.Y.); yanxiaoting1996@163.com (X.Y.); 2School of Pharmaceutical Sciences, Zhejiang University City College, Hangzhou 310000, China

**Keywords:** pharmacometabonomics, tacrolimus, renal transplantation patients, T lymphocytes

## Abstract

The clinical pharmacodynamics of tacrolimus in renal transplant patients has significant interindividual variability. T lymphocytes were selected to study the pharmacodynamic response of tacrolimus, which was significantly correlated with renal function and the outcome of renal transplant patients. Ultra-performance liquid chromatography–quadrupole time-of-flight mass spectroscopy (UPLC/Q-TOF-MS) was performed to obtain the metabolic profiles of 109 renal transplant patients. A partial least squares (PLS) model was constructed to screen potential biomarkers that could predict the efficacy of tacrolimus. Multinomial logistic regression analysis established a bridge that could quantify the relationship between the efficacy of tacrolimus and biomarkers. The results showed a good correlation between endogenous molecules and the efficacy of tacrolimus. Metabolites such as serum creatinine, mesobilirubinogen, L-isoleucine, 5-methoxyindoleacetate, eicosapentaenoic acid, N_2_-succinoylarginine, tryptophyl-arginine, and butyric acid were indicated as candidate biomarkers. In addition, the key biomarkers could correctly predict the efficacy of tacrolimus with an accuracy of 82.5%. Finally, we explored the mechanism of individual variation by pathway analysis, which showed that amino acid metabolism was significantly related to the efficacy of tacrolimus. Moreover, orthogonal partial least squares discriminant analysis (OPLS-DA) showed that there was no difference in key metabolites among different pharmacodynamic groups at 1 month and 3 months after dose adjustment, suggesting that pharmacometabonomics is a useful tool to predict individual differences in pharmacodynamics and thus to facilitate individualized drug therapy.

## 1. Introduction

Delayed graft function and acute rejection are the most common complications after renal transplantation and affect the long-term survival of the patients [1]. According to the data from the Australian and New Zealand Dialysis and Transplant Registry, the incidence of delayed graft function was 24.6%, and the incidence of allograft loss was 33.9% [2]. Tacrolimus (or FK506), a calcineurin inhibitor (CNI), is a first-line treatment option in solid organ transplantation to prevent and treat graft rejection [3,4]. It primarily acts by inhibiting calcineurin dephosphorylation, which is identified as a key signaling enzyme in T cell activation [5]. However, due to the narrow therapeutic window [6] and high variations in the intra- and inter-individual pharmacokinetics of FK506 [7,8], it is hard to formulate an ideal dosage for patients. A higher dosage will contribute to excessive immunosuppression and increase the risk of adverse effects, while a lower dosage will lead to insufficient immunosuppression and increase the risk of rejection [9,10]. Therefore, establishing an effective tool for designing personalized therapy becomes increasingly important. In clinical practice, therapeutic drug monitoring (TDM) is routinely implemented for individualization of FK506 dosage, but blood drug concentration is not necessarily related to pharmacological effects [11]. Moreover, the most frequently used means of therapeutic drug monitoring is to measure the steady-state concentration, which may take some time for patients to reach. Poor control of dosage during this period also has a huge impact on the prognosis. Meanwhile, TDM is a post-event intervention. In addition, genotype testing could also be used to predict the dosage of FK506 [12]; however, patient compliance is poor due to high costs, and the inter-individual variability explained by genotype is less than 40% [13], resulting in a low clinical effective guidance rate.

In addition to genetic variation, environmental factors, including diet, personal physio-pathological conditions, and lifestyles, also play an essential role in drug response [14,15]. Pharmacometabolomics, an emerging “omics” that can obtain the global metabolic profiles of metabolites, is a useful approach to comprehensively analyze the factors of individual variation in drug response [15,16,17]. For instance, complicating toxicity induced by irinotecan has large individual differences and could be predicted using the pre-dose metabolic profile, even with no previous knowledge of the genotype [18]. The pharmacometabonomic approach was also applied to predict the drug response of faropenem based on the pre-dose plasma metabolic features of 36 healthy volunteers by gas chromatography–mass spectrometry [19]. Metabolites had the potential to predict individual variation in the pharmacokinetics (PK) of tacrolimus [16], which focused on predicting pharmacokinetic variations using pre-dose metabolomic data from urine samples. Therefore, metabolic profiling could provide critical information about an individual’s response to a drug, which contributes to personalized medicine.

In this study, we aimed to provide a reference for individualized dosages of tacrolimus after renal transplantation. We recruited 109 renal transplant recipients to investigate the relationship between the pharmacodynamics and metabolic profiling, as well as to identify the most significantly related metabolites for the prediction of the maintenance dosage. In addition, we explored the function of these metabolites in tacrolimus immunosuppression by pathway analysis and enrichment analysis. It provides a reference for individual dose adjustment of tacrolimus in the clinical setting.

## 2. Materials and Methods

### 2.1. Patients

One hundred and nine renal transplant patients were recruited. All participants were enrolled from The First Affiliated Hospital of Zhejiang University between 8 October 2018 and 29 March 2019. They were treated with tacrolimus (FK506), mycophenolate mofetil and corticosteroids. We followed all patients for up to 3 months. Participants were excluded if they died or were under-aged. In addition, we also excluded individuals who were lost to follow-up in the study. The clinical information of participants is provided in Table 1. The study protocol was approved by the Ethics Committee of the First Affiliated Hospital of Zhejiang University, China (2018 llT-938). All participants signed an informed consent form prior to inclusion in this study.

### 2.2. Sample Collection

Fasting blood samples for the determination of trough FK506 concentrations (C0) were taken prior to the morning dose. The collected blood samples were centrifuged at 3000 rpm for 10 min to obtain supernatant plasma. All samples were stored at −80 °C until analysis (avoiding repeated freeze–thaw cycles). Before analysis, 100 μL samples were pretreated by adding 300 μL of acetonitrile, followed by thorough mixing and centrifugation (15 min at 13,000 rpm) in order to completely precipitate the proteins. Pooled quality control (QC) samples were prepared by mixing 10 μL aliquots of plasma from all tested samples, which can be used to evaluate the accuracy and robustness of the method. Next, an aliquot of 100 μL of the filtrates was transferred to autosampler vials for analysis.

### 2.3. Metabolomics Analysis

Non-targeted metabolomics analysis was conducted on a UPLC Q-TOF mass spectrometer (Waters, Manchester, UK) connected to the ACQUITY UPLC system (Waters, Milford, MA, USA) via an electrospray ionization (ESI) interface. Samples were separated on an HSS T3 column (100 mm × 2.1 mm, 1.8 μm) (Waters, Milford, MA, USA). The mobile phase consisted of 0.1% formic acid in water (solvent A) and acetonitrile (solvent B), with a gradient elution of 0–0.5 min, 5% B; 0.5–2 min, 5–20% B; 2–3.5 min, 20–27.5% B; 3.5–4 min, 27.5–70% B; 4–7.5 min, 70–75% B; 7.5–8.5 min, 75–95% B; 8.5–13.5 min, 95% B; 13.5–16 min, 95–5% B; and 16–18 min, 5% B. The flow rate was 0.3 mL/min, and the autosampler was maintained at 10 °C.

The optimal conditions of mass spectrometry were as follows: capillary voltage, 2.5 kV; sample cone, 40 V; source temperature, 100 °C; desolvation temperature, 450 °C; cone gas flow rate, 50 L/h; desolvation gas (N_2_) flow rate, 800 L/h. The ESI source was operated in both positive and negative electrospray ionization (ESI^+/−^) modes. An MS^E^ experiment using two scan functions was carried out as follows: function 1 (low energy): 50–1200 mass-scan range; 0.2 s scan time; 0.015 s inter-scan time; 6 eV collision energy; function 2 (high energy): 50–1200 mass-scan range; 0.2 s scan time; 0.015 s inter-scan time; collision energy ramp of 20–30 eV. Leucine-enkephalin (5 ng/mL) was used as the lock mass for accurate mass acquisition. Additionally, blank samples were run at the beginning of each batch to balance the column and assess the background ions introduced by sample derivatization. Finally, the prepared solutions were kept at 10 °C, and an aliquot (2 μL) of each sample was injected and analyzed by UPLC/Q-TOF-MS in a randomized fashion. During analysis, 1 quality control sample was inserted after every 10 injections of tested samples.

### 2.4. Pharmacodynamic Analysis

The accurate measurement of T lymphocytes was achieved by six-color flow cytometry. The details are as follows: Whole blood was stained with the six-color reagent solution and incubated for 15 min at room temperature in the dark. Then, red blood cells were lysed with erythrocyte lysin and then mixed thoroughly and incubated for 15 min at room temperature in the dark. Finally, flow cytometry was conducted to calculate the number of T lymphocytes. According to the standard of T lymphocyte percentage in our hospital, all patients were divided into three groups: the low-response group, whose T lymphocyte percentage was higher than the upper limit of the reference range (*n* = 16); the normal-response group, whose T lymphocyte percentage was within the reference range (*n* = 73); and the high-response group, whose T lymphocyte percentage was lower than the upper limit of the reference range (*n* = 20).

### 2.5. Statistical Analysis

The obtained raw metabolomics data were analyzed by Progenesis QI ver. 2.2 (Nonlinear Dynamic), including automatic alignment, peak picking, deconvolution, and compound identification. SIMCA-P software (version 14, Umetrics, Umea, Sweden) was employed to analyze the multivariate data matrix. Continuous data are expressed as mean ±  SD. Categorical data were compared by Fisher’s exact test. Correlation analysis was conducted using the Pearson or Spearman rank test, and *p*-value < 0.05 was considered indicative of a statistical significance. Multinomial logistic regression analysis for discriminant analysis was performed using the Statistical Package for the Social Sciences (SPSS, v. 18.0; SPSS, Inc., Chicago, IL, USA).

## 3. Results

### 3.1. Basic Characteristics of the Study Cohorts

The general clinical characteristics and demographic information of all participants on the 7th day after transplantation are presented in Table 1. According to the reasonable range of T lymphocyte percentages in our hospital (53.7–80.9%), patients were divided into three groups. Categorical variables are shown as numbers (%), and continuous variables are shown as means ± standard deviations. There were no differences in age, sex, body mass index (BMI), or liver function among the groups. However, there were significant differences in the renal function indexes, including creatinine, cystatin C, and creatinine clearance rate, which could affect the distribution and elimination process of tacrolimus. Plasma samples were respectively collected from 109 transplant patients in different periods: the 7th day, the 1st month, and the 3rd month after transplantation.

### 3.2. Correlation of T Lymphocytes with Renal Function Indexes and Outcome of Renal Transplantation

The number of T lymphocytes was one of the indicators of the tacrolimus efficacy and played an important role in the prognosis of renal function after transplantation. However, T lymphocytes were highly variable among individuals. Serum creatinine, cystatin C, and glomerular filtration rate were important indexes for evaluating renal function. Pearson correlation analysis was performed, and the results showed that T lymphocytes were significantly correlated with serum creatinine, cystatin C, and glomerular filtration rate (Figure 1). Meanwhile, we found that patients in the high-response group were prone to delayed graft function (*p* < 0.05). Although there was no significant difference in the incidence of acute rejection between the low-response group and the normal group, the risk of acute rejection in the low-response group was much higher than that in the normal group (Table 2). These results indicated that T lymphocytes could be used as an efficacy indicator of tacrolimus in renal transplant patients.

### 3.3. Metabolic Profiling of Plasma Samples

The plasma samples of 109 patients were collected on the 7th day (D7), 1st month (M1), and 3rd month (M3) after transplantation. Under the selected conditions, we detected and identified 2695 metabolites from the LC-MS analysis in ESI^+^ and ESI^-^ modes by searching the HMDB library (https://hmdb.ca/spectra/ms/search, accessed on 18 October 2021) and Lipid Map database (see Appendix A). To ensure data reliability, we preprocessed the data, mainly including deleting metabolites with more than 80% missing values in the samples, using KNN (k-nearest neighbor) to fill in the missing values, retaining metabolites with a relative standard deviation (RSD) in QC of less than 30%, and performing local polynomial fitting on the basis of QC. Coefficients of variation of the distribution of peaks in the QC samples indicated that the present analysis was stable and repeatable (Figure 2). Furthermore, the results of principal component analysis showed that the QC samples clustered tightly together, indicating that the present study was reliable (Figure 2). Finally, 1514 metabolites were selected for further analysis.

### 3.4. Identification of Metabolites Significantly Associated with the Percentage of T Lymphocytes

To select metabolites that were significantly associated with the percentage of T lymphocytes, a two-stage PLS analysis was performed to predict the percentage of T lymphocytes using the metabolites. A scatterplot from the initial PLS analysis is shown in Figure 3, in which each point represents one individual, metabolic profile data are X variables, and the percentage of T lymphocytes, which represent the curative effect of tacrolimus, is the Y variable. The results demonstrated a good correlation, with R^2^ = 0.923 for T cell percentage (Figure 3A). The loading plot of the model is shown in Figure 3B, which shows the correlation between every predictive biomarker and the response variable. The X variables in the top-right corner of Figure 3B are positively correlated with the percentage of T lymphocytes, while the variables in the lower-left corner are negatively correlated with the percentage of T lymphocytes. The variable importance in the projection (VIP) is an index that represents the contribution of X variables to the PLS model. Finally, a set of 95 variables with high VIP values (VIP > 1.5) were applied to predict individual diversity in the initial PLS model.

The 95 selected variables were then used to build the second PLS model. It also showed goodness of fit (R^2^ = 0.709), as well as high predictability (Q^2^ = 0.469) (Figure 3C). Internal validation with 200 permutation tests was conducted to check whether the data were overfitting. The results showed that all R^2^ (goodness-of-fit) and Q^2^ (predictability of model) values from the original model (far right) were larger than those of the permuted models (left), demonstrating the validity of the second PLS model (Figure 3D). Finally, we selected 43 metabolic features with VIP > 1 in the second PLS model to characterize the individualized percentage of T lymphocytes.

Moreover, to further identify the molecules that had a strong correlation with the percentage of T lymphocytes, Pearson correlation analysis was conducted, and the level of 0.05 was defined as significant (*p* < 0.05). Finally, 41 metabolic features were related to the percentage of T lymphocytes, and 19 metabolic features were identified; the results are summarized in Table 3. To investigate the potential functional roles of these metabolites in the individualized percentage of T lymphocytes, we conducted enrichment analysis and pathway analysis using the 19 metabolites with MetaboAnalyst (https://www.metaboanalyst.ca/MetaboAnalyst/home.xhtml, accessed on 12 November 2021). Homo sapiens (KEGG) was selected as the pathway library, and over-representation analysis and pathway topology analysis were based on the hypergeometric test and relative betweenness centrality, respectively. The results showed that the amino acid metabolism pathway was the most relevant to the individualized pharmacodynamic response (Figure 4A,B).

### 3.5. Prediction of the Percentage of T Lymphocytes Based on Key Metabolites and Clinical Characteristics

All individuals were divided into three groups based on the pharmacodynamic response of tacrolimus, and the values of the pharmacodynamic response of tacrolimus were significantly different among the groups. Clinical characteristics were also considered for dosage optimization. Thus, Pearson correlation analysis was conducted to select the related clinical characteristics. The level of serum creatinine had a significant relationship with the T lymphocyte percentage. It was difficult to predict the individual pharmacodynamic response of tacrolimus based on all variables in the clinical setting. To investigate the predictive ability of the above variables, multinomial logistic regression was performed to build a predictive model for pharmacodynamic classification and to further select the variables that had a significant impact on the model. The results revealed that serum creatinine, mesobilirubinogen, L-isoleucine, 5-methoxyindoleacetate, eicosapentaenoic acid, N_2_-succinoylarginine, tryptophyl-arginine, and butyric acid had significant contributions to the model (*p* < 0.05). In order to provide a reference for the individualized dosage of tacrolimus after discharge, daily dosage was included in the model. The detailed information of the model is summarized in Table 4. Based on the results of multinomial logistic regression, the results showed that the prediction accuracy of the model was 82.5%, the overestimation rate was 4.5%, and the underestimation rate was 12.8% (Figure 5A). In the final PLS model, which was built based on nine metabolites, the prediction power (R^2^ = 0.605) (Figure 5B) was 65.5% of that of the initial PLS model, which used 1514 metabolic features, and 85.3% of that of the second PLS model, which used 95 metabolic features, indicating that the prediction capability of the model was reasonably high.

### 3.6. OPLS-DA Models to Characterize Pharmacodynamic Responses

The OPLS-DA model was used to validate the predictive ability of the selected molecules. The OPLS-DA model was built based on the key metabolites and clinical characteristics. Each circle represents an individual in high-response and low-response groups, and score plots show an overview of clustering (Figure 6A). The selected molecules separated the two groups (blue and green dots) clearly; R^2^Y was 0.577, and Q^2^ was 0.445. Meanwhile, permutation tests were performed with 200 iterations to avoid overfitting the predictive models (Figure 6B). The results showed that the values on the left (simulated value) were lower than the values on the right (real value), indicating that the model was stable without the risk of overfitting. It turned out that there were significantly metabolic differences between the high-response group and low-response group. In other words, an integrative approach to analyzing pharmacometabolomic and clinical characteristics may have the ability to characterize individual variances in the pharmacodynamic response phenotype. Consequently, it is expected to potentially avoid adverse drug reactions and to help dosage optimization for each individual. Thus, to validate the practicability of the method, we followed all patients for up to 3 months and collected plasma in the 1st month and 3rd months after transplantation. The OPLS-DA model revealed that there was no difference in the selected metabolites between the high-response group and low-response group (Figure 6C,D).

## 4. Discussion

T lymphocytes are mainly derived from bone marrow, mature and differentiate in the thymus, and play an immunomodulatory role [20]. T lymphocytes are activated by stimulation after renal transplantation, and a large number of cloned and proliferated T lymphocytes will lead to acute rejection [21]. Meanwhile, T lymphocytes also participate in the development of ischemia–reperfusion injury, which is the main factor leading to delayed graft function [22]. In this study, the percentage of T lymphocytes was correlated with the function of the kidney, and individuals with a higher percentage of T lymphocytes were more prone to rejection, while those with a lower percentage of T lymphocytes were more prone to delayed graft function. In this study, more than 30% of individuals’ percentage of T lymphocytes were not within the normal range. Tacrolimus is an immunosuppressive drug that exerts immunosuppression by inhibiting the activation of T cells [23].

Pharmacometabonomics, an approach that has proved effective for personalized medicine, was performed in this study to predict the percentage of T lymphocytes of renal transplantation patients who took tacrolimus as an immunosuppressive drug. Based on pharmacometabonomics analysis, T lymphocytes were found to have a significant correlation with the metabolites, and the amino acid metabolic pathway was the most obvious. Meanwhile, some amino acids have been established as biomarkers for some diseases. In infectious viral diseases, the lack of arginine led to a decrease in T lymphocyte expression, and thus, the virus could not be effectively cleared. However, after supplementation with arginine, the number of T lymphocytes increased, and the virus was also inhibited [24]. In a study of Alzheimer’s disease, it was found that increased concentrations of phenylalanine and isoleucine in the blood could stimulate the differentiation and proliferation of T cells [25]. Meanwhile, isoleucine was significantly correlated with the production of IL-10, which could inhibit the immune response of T cells [26]. Tryptophan metabolism also plays an important role in the immune response. Tryptophan can be metabolized into indole derivatives, induce the production of IL-22 [27], and then promote immune defense and tissue repair [28]. Furthermore, tryptophan can be oxidized to kynurenine through indoleamine 2,3-dioxygenase (IDO), which is related to the immune response [29].

Tacrolimus has been the first-line immunosuppressant for renal transplantation patients, and it has been studied extensively. The first pharmacometabonomics analysis of tacrolimus proposed a prediction equation based on the relationship between metabolites and the parameters of PK [16]. In this study, we focused on the correlation between endogenous molecules and the parameters of PD. Serum creatinine, mesobilirubinogen, l-isoleucine, 5-methoxyindoleacetate, eicosapentaenoic acid, N_2_-succinoylarginine, tryptophyl-arginine, and butyric acid were finally selected as key biomarkers. The level of creatinine in serum was associated with renal ischemia and reperfusion injury [30,31]. Mesobilirubinogen was associated with cirrhosis [32], which might affect the metabolism of tacrolimus. Previous studies have shown that there is a potential relationship between amino acids and T lymphocytes. Some amino acids, such as L-isoleucine and arginine, have been known to be involved in immunosuppressive activities [16] and could affect the response to tacrolimus. 5-Methoxyindoleacetate is the key metabolite of tryptophan [33], which is a biomarker in diagnosing acute kidney injury among tacrolimus-treated kidney transplant patients [34]. Eicosapentaenoic acid improved renal function in CNI-treated transplant recipients by competing with substrates of the COX pathway [35] and acted as an ineffective substrate for vasoconstrictive diene metabolites such as TXA2 [36]. According to a previous study, butyric acid could regulate hyperglycemia caused by tacrolimus [37].

## 5. Conclusions

In this study, we explored the correlation between T lymphocytes and endogenous molecules using the pharmacometabonomics approach. Serum creatinine, mesobilirubinogen, L-isoleucine, 5-methoxyindoleacetate, eicosapentaenoic acid, N2-succinoylarginine, tryptophyl-arginine, and butyric acid were indicated as candidate biomarkers to predict the efficacy of tacrolimus, which showed a high accuracy. The levels of the key metabolites had no difference among different pharmacodynamic groups at 1 month and 3 months after dose adjustment, further validating that the pharmacodynamics approach is beneficial for determining individualized dosages of tacrolimus.

It is worth mentioning that this is the first report that presents correlations between T lymphocytes and endogenous molecules and provides a reference for renal transplantation patients to adjust individualized dosages of tacrolimus, which may be extended to other drugs with a high degree of intra-individual variation. However, the most promising biomarkers chosen in this study should be further validated, and whether the results from this study are very meaningful for other hospitals remains unclear and requires further study.

## Figures and Tables

**Figure 1 molecules-27-03517-f001:**
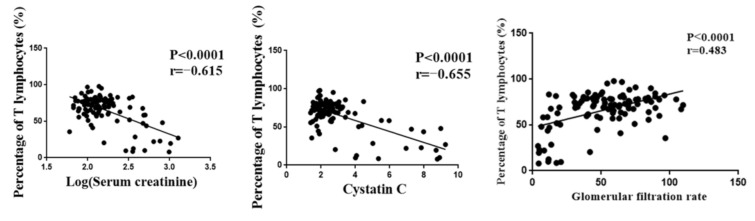
The correlation between T lymphocyte and renal function indexes.

**Figure 2 molecules-27-03517-f002:**
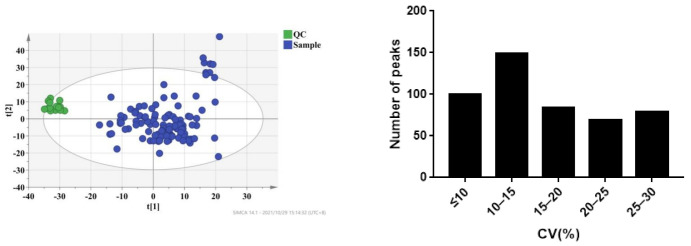
CV distribution of peaks in combinational dataset of ESI^+^ and ESI^−^ modes. After preprocessing the data, all peaks had coefficients of variation (CVs) below 30%.

**Figure 3 molecules-27-03517-f003:**
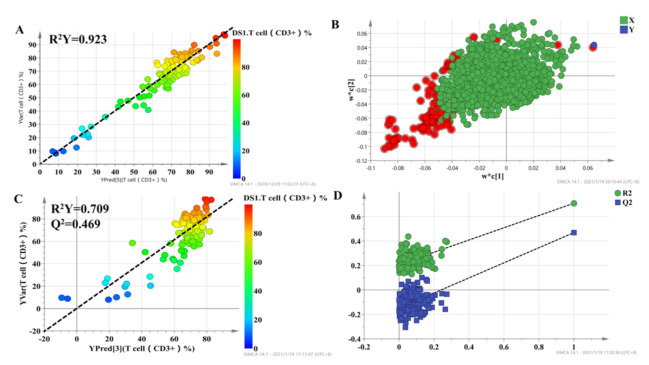
The score plot for the initial PLS model (**A**) and second PLS model (**C**), in which the color from blue to red represents the response variable from low to high. Loading plots for the percentage of T lymphocyte prediction model (**B**), in which the blue box represents the response variable; each circle represents a metabolic feature, and the circle in red represents metabolic features with VIP > 1.5. Internal validation of the second PLS model (**D**) was performed using 200 permutation tests.

**Figure 4 molecules-27-03517-f004:**
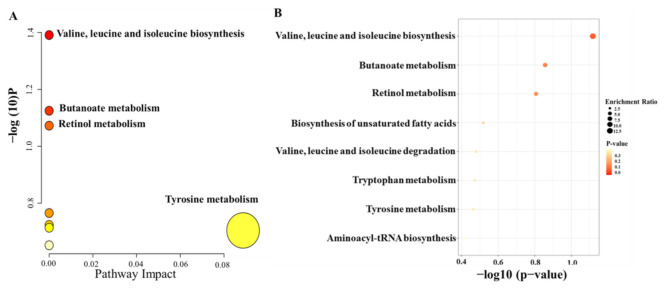
Summary plots of pathway analysis (**A**) and over-representation enrichment analysis (**B**) by MetaboAnalyst (http://www.metaboanalyst.ca, accessed on 12 November 2021) with 19 metabolites selected from the second PLS model.

**Figure 5 molecules-27-03517-f005:**
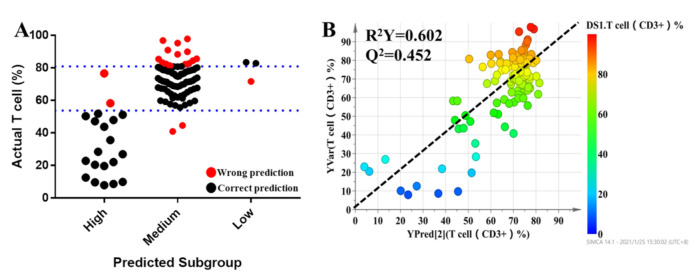
The results of multinomial logistic regression (**A**). Each dot represents an individual. The black dots indicate individuals who could be correctly predicted, while red dots mean individuals who were inaccurately predicted. Refined PLS predicted modeling to predict the percentage of T lymphocytes of tacrolimus (**B**).

**Figure 6 molecules-27-03517-f006:**
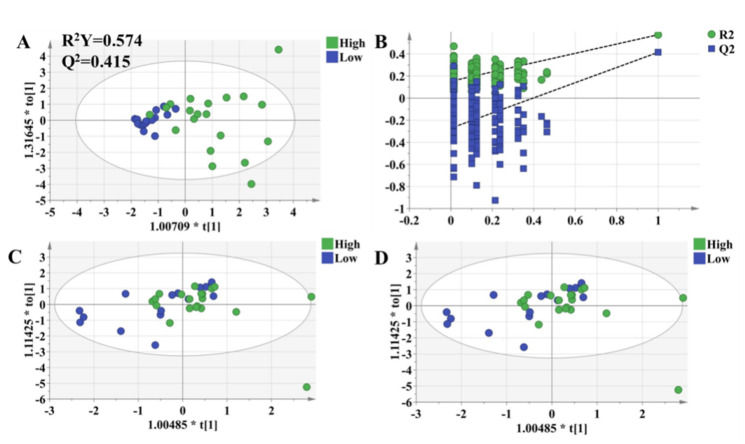
OPLS-DA model to discriminate subgroups based on the selected variables. Each dot represents an individual. Green dots indicate individuals in the high-response group, while blue dots mean individuals in the low-response group. (**A**) The OPLS-DA model of individuals who were 1 week after transplantation. (**B**) Permutation tests of the model in Figure 6A, which were performed with 200 iterations, indicated that the predictive models were not overfitting. (**C**) The OPLS-DA model of the individuals who were 1 month after transplantation. (**D**) The OPLS-DA model of individuals who were 3 months after transplantation.

**Table 1 molecules-27-03517-t001:** Patient characteristics.

Clinical Characteristics	Normal Group (*n* = 73)	Low-Response Group (*n* = 16)	High-Response Group (*n* = 20)	*p*
Ages (years)	40.73 ± 11.29	40.44 ± 11.14	48.8 ± 8.03	0.093
Female (%)	24 (32.9%)	4 (25%)	5 (25%)	0.741
Tacrolimus dose	3.03 ± 0.75	3.19 ± 0.66	2.83 ± 0.90	0.369
BMI (kg/m^2^)	21.46 ± 3.14	21.70 ± 3.18	22.60 ± 3.04	0.254
Cr (μmol/L)	156.43 ± 110.56	154.06 ± 79.36	484.1 ± 338.47	<0.001
CYC (mg/mL)	2.43 ± 0.81	2.38 ± 0.75	5.58 ± 2.71	<0.001
Ccr (ml/min)	57.03 ± 22.62	56.66 ± 23.72	24.13 ± 25.62	<0.001
Albumin (g/L)	36.05 ± 3.04	35.83 ± 3.60	34.99 ± 2.78	0.405
Hemoglobin (g/L)	751.81 ± 538.32	97.35 ± 87.39	1017.98 ± 619.39	<0.001
Hematocrit (%)	30.70 ± 5.77	27.96 ± 5.76	29.53 ± 6.32	0.233
ALT (U/L)	30.23 ± 37.84	35.81 ± 22.31	31.15 ± 26.94	0.843
AST (U/L)	19.22 ± 13.47	18.25 ± 6.23	22.70 ± 16.40	0.532
Total bilirubin (μmol/L)	7.29 ± 3.49	6.73 ± 2.05	8.33 ± 4.94	0.398

Note: Cr, creatinine; CYC, cystatin C; ALT, alanine aminotransferase; AST, aspartate aminotransferase; BUN, blood urea nitrogen; Ccr, creatinine clearance rate.

**Table 2 molecules-27-03517-t002:** The incidence of delayed graft function and acute rejection in different pharmacodynamic subgroups.

Subgroups	Delayed Graft Function (%)	Acute Rejection (%)
Normal group ^a^ (*n* = 73)	3 (4.11%)	7 (9.59%)
Low-response group ^b^ (*n* = 16)	0	4 (25%)
High-response group ^c^ (*n* = 20)	11 (68.75%)	0
*p*	<0.0001	0.105

(*p*: the difference in the incidence of delayed relapse or acute rejection. ^a^ vs. ^b^ or ^a^ vs. ^c^).

**Table 3 molecules-27-03517-t003:** The summary of the 19 identified metabolites.

Metabolites	ID	Adduct (Observed)	Retention Time	m/z (Observed)	VIP	Correlation Coefficient	*p*-Value
Mesobilirubinogen	HMDB01898	[M + H]^+^	4.69	593.33	1.25	−0.337	<0.001
Cinnamoside	HMDB38923	[M + NH_4_]^+^	4.78	536.27	1.23	−0.455	<0.001
L-Isoleucine	HMDB00172	[M + H]^+^	1.05	132.10	1.22	−0.271	0.003
5-Methoxyindoleacetate	HMDB04096	[M − H]^−^	1.04	204.07	1.20	−0.413	<0.001
DG(18:0/22:6(4Z,7Z,10Z,13Z,16Z,19Z)/0:0)	HMDB07179	[M − H]^−^	11.13	667.53	1.19	−0.337	<0.001
PI(16:1(9Z)/0:0)	LMGP06050009	[M + H − H_2_O]^+^	4.71	553.28	1.17	−0.483	<0.001
Isoleucyl-Proline	HMDB03141	[M + H]^+^	1.05	229.15	1.16	−0.500	<0.001
Retinoyl b-glucuronide	HMDB28915	[M + H − H_2_O]^+^	4.64	459.24	1.16	−0.462	<0.001
Tryptophyl-Arginine	HMDB29077	[2M + H]^+^	4.65	721.39	1.15	−0.489	<0.001
Butyric acid	HMDB00039	[2M + H]^+^	4.65	177.11	1.13	−0.480	<0.001
Norepinephrine	HMDB00216	[M + H]^+^	1.05	170.08	1.12	−0.452	<0.001
Eicosapentaenoic acid	HMDB01999	[M − H]^−^	9.72	301.22	1.08	−0.303	0.001
Gamma glutamyl ornithine	HMDB02248	[2M + H]^+^	4.47	523.27	1.05	−0.446	<0.001
Methionyl-Methionine	HMDB28979	[M − H_2_O − H]^-^	3.19	261.07	1.05	−0.309	0.001
Hydroxybutyrylcarnitine	HMDB13127	[M + NH_4_]^+^	4.94	265.18	1.05	−0.359	<0.001
N2-Succinoylarginine	HMDB32764	[2M + H]^+^	4.84	549.26	1.05	−0.299	0.001
L-Aspartyl-L-phenylalanine	HMDB00706	[M - H]^-^	0.98	279.10	1.04	−0.301	0.001
24-Keto-25dehydrocholesterol	LMST01010299	[M + H − H_2_O]^+^	4.93	381.31	1.03	−0.287	0.002
7-Methylhypoxanthine	HMDB03162	[M + H]^+^	1.05	151.06	1.00	−0.373	<0.001

**Table 4 molecules-27-03517-t004:** Parameter estimation of multinomial logistic regression and 95% CI.

Variables	Low-Response Group				High-Response Group			
			95% CI				95% CI	
	*p*-Value	OR	Lower	Upper	*p*-Value	OR	Lower	Upper
Dosages	0.783	0.888	0.383	2.060	0.376	0.555	0.151	2.040
Serum creatinine	0.693	0.471	0.011	19.876	0.104	86.550	0.401	18,686.260
Mesobilirubinogen	0.179	1.135	0.944	1.364	0.010	1.351	1.074	1.699
L-Isoleucine	0.062	0.649	0.412	1.022	0.001	0.256	0.114	0.574
5-Methoxyindoleacetate	0.836	0.934	0.490	1.782	0.040	3.176	1.054	9.568
Eicosapentaenoic acid	0.638	1.082	0.778	1.505	0.019	1.686	1.089	2.608
N_2_-succinoylarginine	0.114	1.014	0.997	1.031	0.108	1.014	0.997	1.031
Tryptophyl-arginine	0.038	1.158	1.008	1.329	0.009	1.357	1.079	1.707
Butyric acid	0.022	0.896	0.816	0.984	0.111	0.889	0.770	1.027

## Data Availability

All data included in this study are available upon request by contact with the corresponding author.

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
