# Peer review of "Individualized Dosage of Tacrolimus for Renal Transplantation Patients Based on Pharmacometabonomics"

_molecules, 2022, doi:10.3390/molecules27113517_

Round 1

Reviewer 1 Report

The manuscript described the metabolic profiling of plasma samples obtained from 109 renal transplant patients. The authors focused on the correlation between endogenous metabolites and the parameters of PD. Finally, serum creatinine, mesobilirubinogen, l-isoleucine, 5-methoxyindoleacetate, eicosapentaenoic acid, N2-succinoylarginine, tryptophyl-arginine, and butyric acid were selected as critical biomarkers. Overall, the study was appropriately designed and implemented. However, there are some issues to concern. Please see the details as follows.

  1. Although some endogenous metabolites were selected, there was no further study to validate the results. A targeted metabolic study should be performed.
  2. The novelty and contribution of this study should be clarified at the end of the Introduction section.
  3. The raw non-targeted metabolomic MS data should be uploaded to a database.
  4. Figure 4b: the size of the text should be increased
  5. Table 3: charge of metabolites should be included.
  6. The authors should use a language editing service to improve style and correct typos and grammar errors.

Reviewer 2 Report

Paper entitled "Individualized dosage of tacrolimus for renal transplantion patients based on pharmacometabonomic" is dealing with contemporary problem of tacrolimus dosing. Tacrolimus is characterised with high variability in pharmacokinetics, and results of this study are adding new data in pool of information regarding patient tailored therapy with tacrolimus. Authors have thorughly examined numerous biomarkers, and after detailed statistical testing identified most promissing biomarkers that should be further studied.

I suggest following elements to be improved before accepting paper for publication

  • title - should it be pharmacometabonomics instead of  pharmacometabonomic
  • conclusion - after very nice storytelling style throughout the entire paper, article itself tends to end ubruptly without proper colnclusion - should be rephrased with couple more sentences added (eg. mentioning again most promissing biomarkers identifieed in the study, that should be further studied)

The main problem is conclusion - last paragraph in discussion section "It is worth to mention that it is the first report presenting the correlation between T lymphocytes and endogenous molecules, and providing a reference for renal tansplantion patients to adjust individualized dosage of tacrolimus, which may be extended to other drugs with high degree of intra-individual variation. However, whether the result from this study is very meaningful for other hospital remains unclear and requires further study."   This is very short conclusion without mentioning any particular data. The conclusion section should contain data mentioned in this part of the abstract.  

Following section has been taken from abstract -  "The metabolites such as serum creatinine, mesobilirubinogen, L-isoleucine, 5-methoxyindoleacetate, eicosapentaenoic acid, N2-succinoylarginine, tryptophyl-arginine, and butyric acid were indicated as candidate biomarkers. In addition, the key biomarkers could well/correctly predict the efficacy of tacrolimus with an accuracy of 82.5 %. Finally, we explored the mechanism of individual variation by pathway analysis, which showed that amino acid metabolism significantly related to the efficacy of tacrolimus. Moreover, orthogonal partial least squares discriminant analysis (OPLS-DA) showed that there was no difference in key metabolites among different pharmacodynamic groups at 1 month and 3 months after dose adjustment, suggesting that pharmacometabonomics was a useful tool to predict individual differences in pharmacodynamicsand thus to facilitate individualized drug therapy."  

Basically, this text should be rephrased (if not copied) and used as conclusion in main text. It should be added before the existing "conclusion" paragraph. 

spell cheking- I always suggest in order to eliminate typos

Round 2

Reviewer 1 Report

The authors appropriately revised the manuscript and resolved all the concerns. The manuscript can be accepted as is for publication in Molecules.